# Rocky Area Inhabiting Daddy Long-Legs Spiders, *Pholcus* Walckenaer, 1805 (Araneae: Pholcidae) in Mountainous Mixed Forests from South Korea

**Chang-Moon Jang** [1,2,3], **Jung-Sun Yoo** [4], **Seung-Tae Kim** [5,*] **and Yang-Seop Bae** [1,2,3,*]

1   Division of Life Sciences, College of Life Sciences and Bioengineering, Incheon National University, Incheon 22012, Republic of Korea
2   Convergence Research Center for Insect Vectors, Incheon 22012, Republic of Korea
3   Bio-Resource and Environmental Center, Division of Life Sciences, College of Life Sciences and Bioengineering, Incheon National University, Songdo-dong, Incheon 22012, Republic of Korea
4   Species Diversity Research Division, National Institute of Biological Resources, Incheon 22689, Republic of Korea
5   Life and Environment Research Institute, Konkuk University, Seoul 05029, Republic of Korea
*   Correspondence: stkim2000@hanmail.net (S.-T.K.); baeys@inu.ac.kr (Y.-S.B.)

**Abstract:** Two new spider species of the genus *Pholcus* Walckenaer, 1805, *Pholcus deokjeok* **sp. nov.** and *Pholcus gangneung* **sp. nov.** in the family Pholcidae C. L. Koch, 1850 are newly described from South Korea. The present new species belong to the *phungiformes* group in the genus. They are found on rock walls in mountainous mixed forests. This work provides diagnoses, detailed descriptions, and taxonomic photographs for these new species. The unusual shaped and strongly sclerotized embolus of *P. gangneung* **sp. nov.** in the *Pholcus phungiformes* group is the first to be reported.

**Keywords:** diagnosis; habitat; new species; *phungiformes* group; rock wall; taxonomy





## 1. Introduction

Pholcidae C. L. Koch, 1850 is one of the most diversified and largest families comprising 1836 species in 97 genera within the order Araneae Clerck, 1757 [1]. To date, 43 species of pholcid spiders have been described in 3 genera from various ecosystems in Korea [1–9]. Pholcidae is one of the taxa that has not been explored much yet in Korea. The genus *Pholcus* Walckenaer, 1805 among the largest genera in the family is known to mainly thrive on dusky, humid spaces such as rock walls and road drains in mountainous regions [8,10]. As of 2021, Korea's forest area was 6298 ha, accounting for 62.7% of the total national area; most of them are mountainous areas composed of conifer and deciduous mixed forests [11], and most mountainous forests have many rock walls suitable for the habitat of *Pholcus* spiders. The genus *Pholcus* can be distinguished from other genera by the combination of the male chelicerae with a pair of proximal frontal apophyses and epigynum sclerotized with a knob [12]. The *Pholcus phungiformes* group can be distinguished from other species group by the male chelicera with frontal apophysis, male palpal tibia with prolatero-ventral modification, male genital bulb without an appendix or having a pseudoappendix, dorsal spine on the procursus, and epigynum sclerotized with a knob [9,10,12]. In Korea, 37 *Pholcus* species belonging to this species group have been described [9]. Two new spiders belonging to the *P. phungiformes* group were collected during a seasonal survey on the spider fauna of mountainous mixed forests in 2022 (Figure 1) and are described with measurements, illustrations, and a diagnosis.

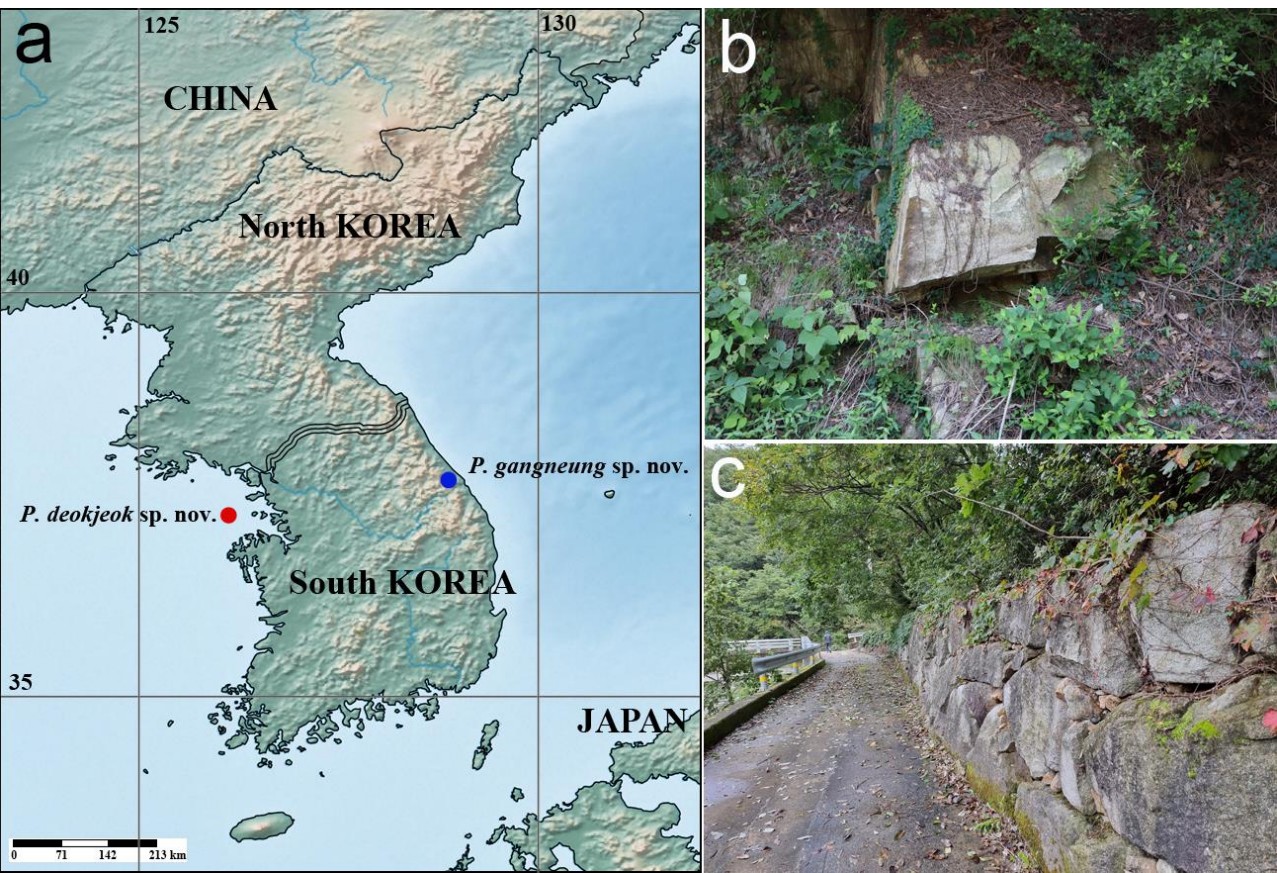

**Figure 1.** Distribution map and habitats of new *Pholcus* species. (**a**) Distribution of the new *Pholcus* in South Korea. (**b**) Habitat of *Pholcus deokjeok* **sp. nov.** (**c**) Habitat of *Pholcus deokjeok* **sp. nov**.

## 2. Materials and Methods

All specimens were preserved in 98% ethyl alcohol, and external morphology was examined under a Leica S8APO (Leica, Singapore) stereomicroscope. Images were captured with a Tucsen Dhyana 400DC digital camera (Fuzhou Tucsen Photonics Co., Ltd., Fuzhou, China) mounted on a Leica S8APO and assembled using Helicon Focus 8.2.0 image stacking software [13]. The female epigynum was dissected and cleared in 10% KOH for 2 h to examine the internal genitalia before illustration. Leg measurements are shown as follows: total length (femur, patella, tibia, metatarsus, tarsus). The morphological terminology follows Huber [12]. The holotype specimens studied are deposited in the Animal Resources Division of the National Institute of Biological Resources, Incheon (NIBR), and the paratype specimens are deposited in the Life and Environment Research Institute of Konkuk University, Seoul (KKU), South Korea. The distribution map was produced by modifying SimpleMappr [14]. The following abbreviations are used in the descriptions: ALE = anterior lateral eye; AME = anterior median eye; PLE = posterior lateral eye; PME = posterior median eye; ALE–AME = distance between ALE–AME; ALE–PME = distance between ALE–PME; AME–AME = distance between AMEs; AME–PME = distance between AME–PME; PLE–PME = distance between PLE–PME; PME–PME = distance between PMEs in the eye region; L/d = length/diameter in the leg measurement.

## 3. Results

**Taxonomic accounts** (Figures 2 and 3)
**Family Pholcidae C. L. Koch, 1850**
**Genus *Pholcus* Walckenaer, 1805**
***Pholcus phungiformes* species group**

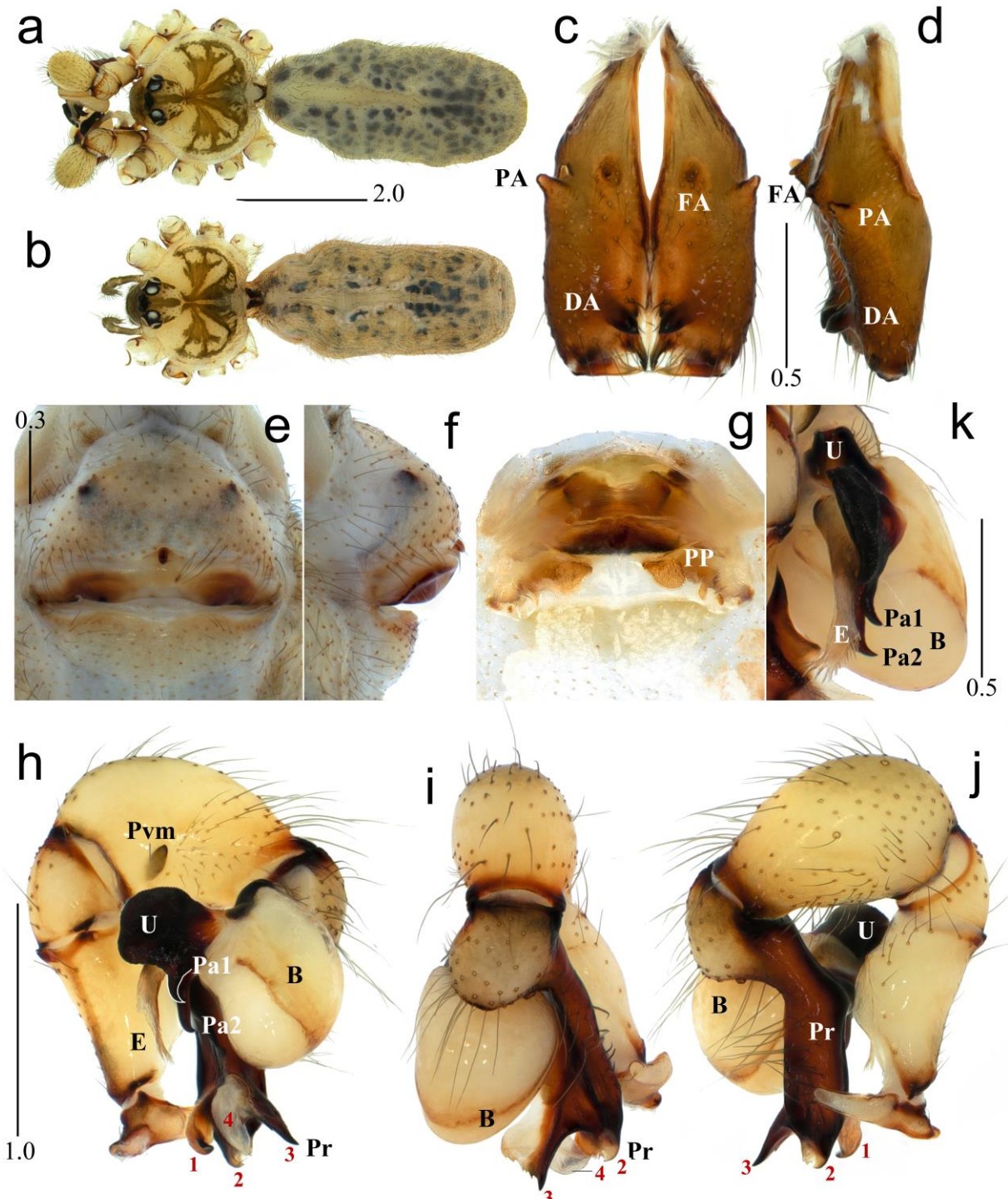

**Figure 2.** *Pholcus deokjeok* **sp. nov.** (**a**) Holotype male (Habitus). (**b**) Paratype female (Habitus). (**c**) Male chelicerae, frontal view. (**d**) *Ditto*, lateral view. (**e**) Female epigynum, ventral view. (**f**) *Ditto*, lateral view. (**g**) Female internal genitalia, dorsal view. (**h**) Male palp, prolateral view. (**i**) *Ditto*, frontal view. (**j**) *Ditto*, retrolateral view. (**k**) Embolic division (1 = ventral apophysis, 2 = retrolateral apophysis, 3 = dorsal apophysis, 4 = prolateral apophysis, B = bulb, DA = distal apophysis, E = embolus, FA = frontal apophysis, PA = proximo-lateral apophysis, Pa = pseudoappendix, PP = pore plate, Pr = procursus, Pvm = prolatero-ventral modification, U = uncus). Scale bars in mm.

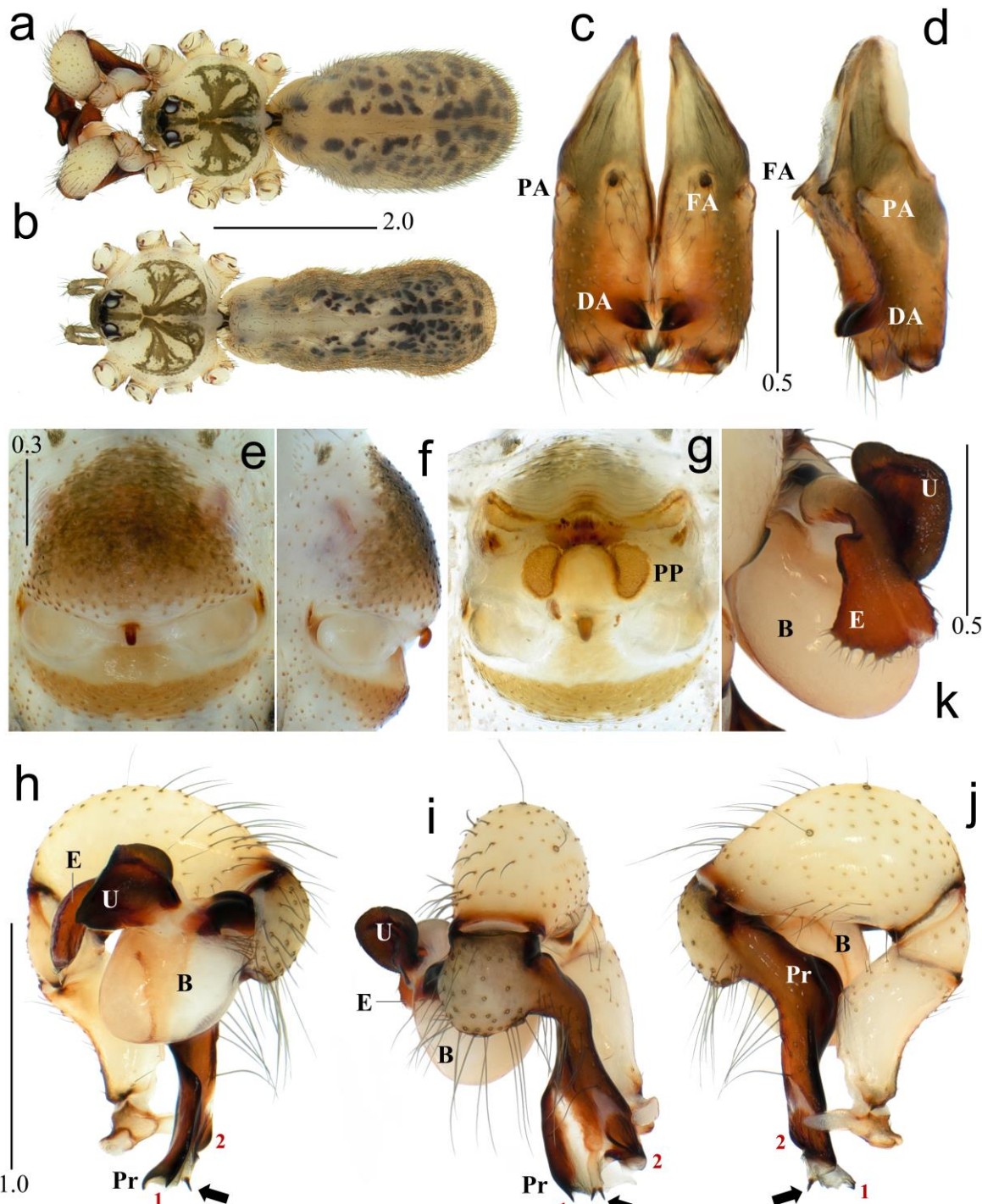

**Figure 3.** *Pholcus gangneung* **sp. nov.** (**a**) Holotype male (Habitus). (**b**) Paratype female (Habitus). (**c**) Male chelicerae, frontal view. (**d**) *Ditto*, lateral view. (**e**) Female epigynum, ventral view. (**f**) *Ditto*, lateral view. (**g**) Female internal genitalia, dorsal view. (**h**) Male palp, prolateral view. (**i**) *Ditto*, frontal view. (**j**) *Ditto*, retrolateral view. (**k**) Embolic division (1 = prolatero-ventral apophysis, 2 = retrolateral apophysis, B = bulb, DA = distal apophysis, E = embolus, FA = frontal apophysis, PA = proximo-lateral apophysis, PP = pore plate, Pr = procursus, U = uncus). Scale bars in mm.

### 3.1. *Pholcus deokjeok* sp. nov.

(Figure 1a,b and Figure 2)

**Type material.** Holotype: ♂, Bijobong Peak, Deokjeokdo Island, Jin-ri, Deokjeok-myeon, Ongjin-gun, Incheon-si, South Korea (37.214236N, 126.124386E, alt. 65 m), 26

July 2022, C.M. Jang leg. (NIBR, #WGJTIV0000000568). **Paratypes:** 6♀♀, same data as the holotype (KKU, #Ara_Phol_Deokjeok_202201~06_PT).

**Etymology.** The specific name is a noun in apposition and refers to the type locality, Deokjeokdo Island.

**Diagnosis.** The new species is similar to *Pholcus chiakensis* Seo, 2014 in the shape of the palpal organ and body appearance but can be easily distinguished from the latter by the combination of the following characters: Male—uncus with two short and long hook-shaped pseudoappendices (Figure 2h,k); procursus with four distal apophyses (two thick ventral and retrolateral apophyses with hooked tips numbered 1 and 2 in Figure 2h–j; one spear-shaped and broad dorsal apophysis with appointed numbered 3 in Figure 2h–j; one thick, finger-shaped, and membranous prolateral apophysis with a pointed tip numbered 4 in Figure 2h,i) versus uncus with two slender and knife-shaped, and pointed short pseudoappendices; procursus with four distal apophyses (one small pointed ventral apophysis; one retrolateral apophysis with a hooked tip; one spear-shaped and broad dorsal apophysis with a twisted tip; and one thick, irregular-shaped, and membranous prolateral apophysis with a blunt tip) in *P. chiakensis* (Figure 2A–C) [15]. Female—a pair of semicircle-shaped pore plates and projected inward in the internal genitalia (Figure 2e–g) versus a pair of European pear-shaped pore plates in the internal genitalia in *P. chiakensis* (Figure 2G,H) [15].

**Description. Male (holotype).** Habitus as in Figure 2a. Total length 5.56. Carapace: 1.84 long/1.95 wide. Eyes: AER 0.70, PER 0.74, ALE 0.18, AME 0.11, PLE 0.18, PME 0.16, ALE–PLE contiguous, AME–ALE 0.06, AME–AME 0.08, AME–PME 0.05, PME–PLE 0.04, PME–PME 0.30. Chelicera: 1.18 long/0.34 wide. Endite: 0.58 long/0.35 wide. Labium: 0.30 long/0.39 wide. Sternum: 0.90 long/1.19 wide. Legs: I 48.15 (12.19, 0.71, 12.62, 20.27, 2.36)/II 31.78 (8.96, 0.68, 8.02, 12.64, 1.48)/III 22.51 (6.46, 0.68, 5.55, 8.54, 1.28)/IV 28.19 (8.16, 0.64, 7.16, 11.04, 1.19), tibia I L/d 83. Palp: 3.71 (0.78, 0.41, 1.15, -, 1.37). Abdomen: 3.72 long/1.68 wide.

Carapace pale yellowish brown, cephalic region with pale blackish brown median and marginal bands, thoracic region with pale blackish brown radial and marginal bands (Figure 2a). Chelicera with three apophyses; blunt proximo-lateral apophysis diagonally upward and protrudent out of chelicera, small and pointed frontal apophysis protrudent forward, and thick and pointed distal apophysis diagonally downward (Figure 2c,d). Legs yellowish brown, retrolateral trichobothrium on tibia I at 4% proximally, tarsus I with >40 pseudosegments (only distally about 10 visible), 3/4 of femora brown or pale blackish brown proximally with one brown distal annulus, tibia I with one brown annulus at proximal end and two distal annuli, tibiae II–IV with two brown proximal annuli and two distal annuli, metatarsi with one brown annulus at proximal end, leg formula I–II–IV–III. Abdomen elliptical, pale greenish yellow with a long cardiac pattern and many black irregular spots (Figure 2a). Palp (Figure 2h–k): trochanter with blunt and bulged retrolatero-ventral apophysis, shorter than femur; palpal tibia with finger-shaped prolatero-ventral modification (Figure 2h); bulb pale yellowish brown, cordiform, appendix absent (Figure 2h); uncus dark blackish brown and fist-shaped with fine scales, edge finely serrated, two hook-shaped pseudoappendices present, pseudoappendix 1 shorter than pseudoappendix 2 (Figure 2h,k); embolus unmodified, weakly sclerotized with some semi-transparent distal fringed processes (Figure 2h,k); procursus large and brown with dark blackish brown margin, distinct ventral knee present, four distal apophyses present, two thick ventral and retrolateral apophyses with hooked tips (numbered 1, 2 in Figure 2h–j); one spear-shaped (in lateral view) and broad (in frontal view) dorsal apophysis partly sclerotized and membranous with many distal processes (numbered 3 in Figure 2h–j); one thick, finger-shaped, and membranous prolateral apophysis with a pointed tip (numbered 4 in Figure 2h,i); dorsal spine absent.

**Female.** General appearance similar to male, habitus as in Figure 2b. Total length 4.92. Carapace: 1.63 long/1.88 wide. Eyes: AER 0.64, PER 0.70, ALE 0.17, AME 0.12, PLE 0.18, PME 0.15, ALE–PLE contiguous, AME–ALE 0.07, AME–AME 0.05, AME–PME 0.06, PME–PLE 0.06, PME–PME 0.23. Chelicera: 1.01 long/0.32 wide. Endite: 0.55 long/0.29

wide. Labium: 0.24 long/0.37 wide. Sternum: 0.87 long/1.14 wide. Legs: I 35.25 (8.88, 0.69, 8.98, 14.54, 2.16)/II 28.35 (8.04, 0.63, 7.15, 11.24, 1.29)/III 17.31 (5.02, 0.57, 4.24, 6.54, 0.94)/IV 24.08 (7.01, 0.58, 6.05, 9.25, 1.19), tibia I L/d 60. Palp: 1.17 (0.38, 0.17, 0.21, -, 0.41). Abdomen: 3.29 long/1.58 wide. Epigynum: 1.14 wide.

Legs yellowish brown, femora with two pale brown proximal annuli and two brown distal annuli, tibiae with two brown proximal annuli and two brown distal annuli, metatarsi with one brown annulus at proximal end, leg formula I–II–IV–III. Epigynum (Figure 2e,f): sclerotized, anterior epigynal plate strongly protrudent, anterior arch with median portion almost straight; small and short knob with a blunt tip. Internal genitalia (Figure 2g): anterior arch straight with a pair of semicircle-shaped pore plates and projected inward.

**Habitat.** The species was collected by hand on rock walls and under rocks in a hilly mixed forest (Figure 1b).

**Distribution.** South Korea (Deokjeokdo Island, Incheon-si) (Figure 1a).

*3.2. Pholcus gangneung sp. nov.*

(Figure 1a,c and Figure 3)

**Type material. Holotype:** ♂, Mt. Jabyeongsan, Sangye-ri, Okgye-myeon, Gangneung-si, Gangwon-do, South Korea (37.573733N, 128.961348E, alt. 119 m), 19 September 2022, C.M. Jang & S.T. Kim leg. (NIBR, #NIBRIV0000901598). **Paratypes:** 11♀♀4♂♂, same data as the holotype (KKU, #Ara_Phol_Gangneung_202201~15_PT).

**Etymology.** The specific name is a noun in apposition and refers to the type locality, Gangneung-si.

**Diagnosis.** This species can be distinguished from the other *Pholcus phungiformes* group members by the combination of the following characters: Male—embolus highly modified, sclerotized with 16–17 distal spiny process (Figure 3h,k); procursus with two simple distal apophyses present (one bifurcate and broad prolatero-ventral apophysis with pointed corners numbered 1 in Figure 3h–j, one round retrolateral apophysis numbered 2 in Figure 3h–j); two ventral spines present at the distal end (arrowed in Figure 3h–j). Female—epigynal anterior arch with median portion procurved, a small knob with a blunt tip curved downward; anterior arch chevron-shaped with a pair of kidney-shaped pore plates in the internal genitalia (Figure 3e–g). The unusual shaped and strongly sclerotized embolus of *P. gangneung* **sp. nov.** in the *Pholcus phungiformes* group is the first to be reported.

**Description. Male (holotype).** Habitus as in Figure 3a. Total length 4.49. Carapace: 1.54 long/1.62 wide. Eyes: AER 0.71, PER 0.77, ALE 0.18, AME 0.15, PLE 0.19, PME 0.18, ALE–PLE contiguous, AME–ALE 0.05, AME–AME 0.07, AME–PME 0.05, PME–PLE 0.05, PME–PME 0.29. Chelicera: 1.05 long/0.30 wide. Endite: 0.47 long/0.29 wide. Labium: 0.28 long/0.37 wide. Sternum: 0.84 long/1.11 wide. Legs: I 37.78 (9.71, 0.61, 9.75, 15.57, 2.14)/II 28.45 (7.32, 0.67, 6.63, 12.50, 1.33)/III 19.72 (5.36, 0.54, 4.54, 8.22, 1.06)/IV 25.55 (6.66, 0.60, 6.01, 11.02, 1.26), tibia I L/d 66. Palp: 3.39 (0.66, 0.38, 1.01, -, 1.34). Abdomen: 2.95 long/1.71 wide.

Carapace pale yellowish brown, cephalic region with pale blackish brown median and marginal bands, thoracic region with pale blackish brown radial and marginal bands (Figure 3a). Chelicera with three apophyses; blunt proximo-lateral apophysis diagonally upward and unprotrudent out of chelicera, small and pointed frontal apophysis protrudent forward, and thick and pointed distal apophysis slightly diagonally downward (Figure 3c,d). Legs pale yellowish brown, retrolateral trichobothrium on tibia I at 4% proximally, tarsus I with >35 pseudosegments (only distally about 10 visible), femora with two faint proximal annuli and two blackish brown distal annuli, tibia I with one blackish brown proximal annulus and one distal annulus, tibiae II–IV with two blackish brown proximal annuli and two distal annuli, metatarsi with one blackish brown annulus at proximal end, leg formula I–II–IV–III. Abdomen elliptical, pale yellowish red with a long cardiac pattern and many black irregular spots (Figure 3a). Palp (Figure 3h–k): trochanter with blunt and finger-like retrolatero-ventral apophysis, shorter than femur; palpal tibia with finger-shaped prolatero-ventral modification hidden by uncus; bulb pale yellowish brown,

pocket-shaped, appendix absent (Figure 3h); uncus dark blackish brown and fist-shaped with a truncated edge and fine scales, edge finely serrated, pseudoappendix absent; embolus strongly modified, brown, thick, sclerotized, and elongated elliptical in prolateral view, broad spatula-shaped with 16–17 spiny processes distally such as a cactus in frontal view (Figure 3h,k); procursus large, simple, and brown with dark blackish brown margin, distinct ventral knee present, two distal apophyses present, one bifurcate (in lateral view) and broad (in frontal view) prolatero-ventral apophysis partly sclerotized and membranous with pointed corners (numbered 1 in Figure 3h–j), one round retrolateral apophysis (numbered 2 in Figure 3h–j); dorsal spine absent; two ventral spines present at distal end (arrowed in Figure 3h–j).

**Female.** General appearance similar to male, habitus as in Figure 3b. Total length 4.89. Carapace: 1.57 long/1.72 wide. Eyes: AER 0.59, PER 0.66, ALE 0.17, AME 0.11, PLE 0.15, PME 0.16, ALE–PLE contiguous, AME–ALE 0.03, AME–AME 0.06, AME–PME 0.04, PME–PLE 0.06, PME–PME 0.22. Chelicera: 0.94 long/0.31 wide. Endite: 0.47 long/0.28 wide. Labium: 0.26 long/0.34 wide. Sternum: 0.78 long/1.04 wide. Legs: I 33.62 (8.47, 0.66, 8.48, 13.84, 2.17)/II 23.06 (6.24, 0.65, 5.75, 9.04, 1.38)/III 16.56 (4.71, 0.60, 3.98, 6.20, 1.07)/IV 22.3 (6.35, 0.60, 5.40, 8.58, 1.37), tibia I L/d 57. Palp: 1.39 (0.39, 0.17, 0.26, -, 0.45). Abdomen: 3.32 long/1.44 wide. Epigynum: 1.12 wide.

Legs yellowish brown, femora with two pale brown proximal annuli and two blackish brown distal annuli, tibiae with two brown proximal annuli and two brown distal annuli, metatarsi with a blackish brown annulus at the proximal end, leg formula I–II–IV–III. Epigynum (Figure 3e,f): anterior epigynal plate slightly protrudent, anterior arch with median portion procurved; a small knob with a blunt tip curved downward. Internal genitalia (Figure 3g): sclerotized, anterior arch chevron-shaped with a pair of kidney-shaped pore plates.

**Habitat.** The species was collected by hand on artificially constructed rock walls in a mountainous mixed forest (Figure 1c).

**Distribution.** South Korea (Gangneung-si, Gangwon-do) (Figure 1a).

## 4. Discussion

This study taxonomically describes two new species belonging to the *Pholcus phungiformes* group in the genus *Pholcus* and is considered to be an important contribution to understand the Korean spider fauna. Huber described that the *P. phungiformes* group is distinguished from other species groups by the combination of the following diagnostic characters: male chelicerae with proximo-frontal apophyses (absent in *P. beijingensis*), male palpal tibia with prolatero-ventral modification, procursus with dorsal spines (absent in *P. alloctospilus*, *P. beijingensis*, *P. chiakensis*, *P. palgongensis*, *P. piagolensis*, *P. uiseongensis*, and *P. yeongwol*), appendix absent, sometimes with a pseudoappendix, and epigynum sclerotized with a knob [12,15–18]. Of the newly described spiders in the present study, *P. gangneung* **sp. nov.** has proximo-frontal apophyses on male chelicerae (Figure 3c,d), prolatero-ventral modification on male palpal tibia, and sclerotized epigynum with a knob (Figure 3e,f). In view of these diagnostic characters, *P. gangneung* **sp. nov.** is considered to belong to the *P. phungiformes* group. However, Huber described that the embolus in this species group is weakly sclerotized [12], while that of *P. gangneung* **sp. nov.** is strongly sclerotized (Figure 3h,k). Nevertheless, we tentatively placed this new species in the *P. phungiformes* group instead of erecting a new species group only with a degree of sclerotization because the diagnostic characters of *P. gangneung* **sp. nov.** are still close to this species group in the current taxonomic status. The *P. phungiformes* group is largely restricted to northeastern China, the Korean Peninsula, and Russia (Far East) [12,18]. Recently, taxonomic studies belonging to this species group have been actively conducted in Korea and China, and many species have been newly described in the past decade [8–10,16–18]. Despite many recent studies on this species group, Pholcidae is still one of the taxa that has not been explored much yet, and many species that have not yet been taxonomically studied appear to be waiting for new names in Korea. Further studies will be needed to

provide more definitive answers to the difference of the embolus of *P. gangneung* **sp. nov.** within the *P. phungiformes* group.

The *phungiformes* species in the genus *Pholcus* are mostly found around rocky areas in mountainous mixed forests and is therefore an important natural enemy of forest insect pests flying around such environments. In addition to exploring the species that have not yet been discovered, it would also be important to accumulate further biological information of the *Pholcus* spiders for the understanding of the various ecological roles they play within their ecosystems.

**Author Contributions:** C.-M.J.: conceptualization, methodology, investigation, collection, identification, and original draft preparation. S.-T.K.: conceptualization, methodology, investigation, collection, identification, original draft preparation, project administration, and funding acquisition. J.-S.Y.: conceptualization, review and editing, and funding acquisition. Y.-S.B.: conceptualization, methodology, and review and editing. All authors have read and agreed to the published version of the manuscript.

**Funding:** This work was supported by grants from the National Institute of Biological Resources (NIBR) funded by the Ministry of Environment (MOE) (NIBR202203105 and 202203202), the National Research Foundation of Korea (NRF) funded by the Ministry of Education (2020R1A6A1A03041954), and the R&D Program for Forest Science Technology (2017042B10-2223-CA01) funded by Korea Forest Service (Korea Forestry Promotion Institute) of the Republic of Korea.

**Data Availability Statement:** Not applicable.

**Acknowledgments:** We are grateful to Sue Yeon Lee (Life and Environment Research Institute, Konkuk University) and Ulziijargal Bayarsaikhan (Division of Life Sciences, College of Life Sciences and Bioengineering, Incheon National University) for contributing to the fieldwork and valuable suggestions with critical comments on the original draft.

**Conflicts of Interest:** The authors declare no conflict of interest.

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
