# Peer review of "Rocky Area Inhabiting Daddy Long-Legs Spiders, Pholcus Walckenaer, 1805 (Araneae: Pholcidae) in Mountainous Mixed Forests from South Korea"

_forests, doi:10.3390/f14030538_

Round 1
Reviewer 1 Report
I have only minor comments marked directly in the attached file. In my opinion the manuscript is acceptable after minor revision.

Author Response
Dear Reviewer,
We checked your comments very carefully and revised the manuscript considering them. Your suggestions were heeded and corrected in the text.
Please see the the file we sent you.
In the revised paper, the text related to you (review1) was changed to blue.
We appreciate you for consideration and sincere advice for our manuscript again and hope to hear a good result from you.
Thanks.

Reviewer 2 Report
This is nice taxonomical study, however It is not showing any relation-connection to journal of Forests, which really should be somehow connected to forest ecosystems. It is just taxonomical study on two pholcid spiders, therefore it is really more suitable to another journals which clear taxonomical focus such as Zootaxa, Zookeys etc. Therefore, I am suggesting reject the MS and sending it to another journal.
Author Response
Dear Reviewer,
We consider that the study on two pholcid spiders is related to journal of Forests.
We appreciate your consideration and time.
Sinderely
Reviewer 3 Report
Dear Editor and authors,
The manuscript "Rocky area inhabiting daddy long-legs spiders, Pholcus 2 Walckenaer, 1805 (Araneae: Pholcidae) in mountainous mixed 3 forests from South Korea" is a high-quality taxonomic paper, in which two new species are described from Korea.
In the manuscript PDF, I have detailed minor questions that deserves further revision by the authors. However, three main questions require attention:
1. The use of the term "mixed forest" is not clear to me. Does this reffer to a specific type of forest in Korea? Or does this reffers to a secondary vegetation? If it is a secondary vegetation, than the introduction must not emphasize much the biodiversity shortfalls of Korean forests, as the newly described species were not even sampled at primary forests in the country.
2. The authors provided a general discussion, that could be removed without loosing any important information. State that spiders eat insects and that newly described species are important in ecossystems as predators of insect pests is not relevant at all. Besides, this is not an ecological study and the authors has no further information on the natural history and ecology of the newly described species.
3. The one thing in this manuscript that really deserves to be discussed is the inclusion of Pholcus gangneung in the phungiformes species-group. I do not see any reason to this inclusion. However, the authors merely stated that this species is highly modified, without any support for their assumption that this species belongs to this species-group. This species does not match the definiton of this species-groupd by Huber (2011).

Author Response
Dear Reviewer,
We checked your comments very carefully and revised the manuscript considering them. Your suggestions were heeded and corrected in the text.
Please see the the file we sent you.
In the revised paper, the text related to you (review1) was changed to red.
We appreciate you for consideration and sincere advice for our manuscript again and hope to hear a good result from you.
Thanks.

Round 2
Reviewer 2 Report
I still think that this MS is more suitable to clasicall taxonomical journal than Forests. You can see description: Forests is a peer-reviewed, open access journal of forestry and forest ecology published monthly online by MDPI.
How it is related to Forestry and forest ecology? It is just two records of new species located in rocky habitat of forest, without any discussion on ecological affinities and relation in focus of forest ecology. Therefore, I will not make any new revision, and I hope that editorial office of this journal will not send me similar MS again for revision. The decision is now just up to the editor. Thanks.